# Alpha Spectrometry of Radon Short-Lived Progeny in Drinking Water and Assessment of the Public Effective Dose: Results from the Cilento Area, Province of Salerno, Southern Italy

**DOI:** 10.3390/ma13245840

**Published:** 2020-12-21

**Authors:** Enver Faella, Simona Mancini, Michele Guida, Albina Cuomo, Domenico Guida

**Affiliations:** 1Department of Physics (DF), University of Salerno, 84084 Fisciano, Italy; efaella@unisa.it; 2Laboratory “Ambients and Radiations” (Amb.Ra.), DIEM, University of Salerno, 84084 Fisciano, Italy; smancini@unisa.it; 3Department of Computer Engineering, Electrical Engineering and Applied Mathematics (DIEM), University of Salerno, 84084 Fisciano, Italy; 4Department of Civil Engineering (DICIV), University of Salerno, 84084 Fisciano, Italy; acuomo@unisa.it (A.C.); dguida@unisa.it (D.G.); 5InterUniversity Center for the Prediction and Prevention of Major Hazards (C.U.G.RI.), University of Salerno, 84084 Fisciano, Italy

**Keywords:** radon, radon in drinking water, progeny, ionization chamber, solid-state detector, alpha spectrometry, ingestion, effective dose, public health hazard

## Abstract

Radon is a naturally occurring radioactive gas present in the hydrosphere, lithosphere and atmosphere abundantly. Its ionizing radiation provides the largest human internal exposure by inhalation and ingestion to natural sources, constituting a serious health hazard. The contribution to total exposure is mainly due to inhalation, as ingestion by food or drinking water is typically very small. However, because of public health concerns, the contributions from all these sources are limited by regulations and remedial action should be taken in the event that the defined threshold values are overcome. In this paper, the first campaign of measurements to control the radon activity concentration in drinking water from public water supplies in the province of Salerno, south Italy, is described. The results represent a main reference for the area, as it was never investigated before. The purpose of this survey was to contribute to data compilation concerning the presence of radon-222 in groundwater in the Campania region and to determine the associated risk for different age groups. The maximum radon activity concentrations and the related total annual public effective dose turned out to be lower than the threshold values (100 Bq/l and 0.1 mSv/y, respectively) indicated by international guidelines and the national regulation, showing that the health risks for public consumption can be considered negligible.

## 1. Introduction

Safe drinking water must “not represent any significant risk to health over a lifetime consumption, including different sensitivities that may occur between life stages” [1]. Water-quality assurance is one of the most important issues in environmental programs in order to protect public health. Drinking water as well as air, soil, food, etc. naturally contain terrestrial radioactive elements. Therefore, a radiological public health hazard may be derived from the ionizing radiation emitted. The occurrence of radionuclides in drinking water, in fact, gives rise to direct internal exposure through ingestion and inhalation. Therefore, measurement of radioactivity in drinking water is important in order to determine the extent of exposure of the population to radiation due to habitual daily consumption of water. A very high level of radon in drinking water can lead, in the long term, to a significant risk of developing internal organ diseases, primarily stomach and gastrointestinal cancer.

For such a reason, ^222^Rn, derived from the alpha decay of ^226^Ra, is the most relevant one from a radioprotection point of view.

Because of its long half-life (3.82 d) and its aptitude for the formation of chemical bonds, its diffusion in the environment is easily favoured. In fact, after production, Radon is able to migrate inside the rock body, where it is generated through pores and fractures transported by the fluid carriers present in the subsoil. When the pores are saturated with water, radon dissolves in it and is transported by it. Therefore, the concentration of radon activity in groundwater is strictly correlated with local geology, derived from direct emission by the rocks forming the aquifer.

However, as radon can be easily desorbed because of the water/air partition coefficient as a function of temperature [2,3] favourable to the air, activity concentrations are typically higher in groundwater than in surface waters.

For such a reason, the potabilization and purification treatments carried out on water to make it usable for drinking purposes usually not only improve its quality but also reduce its radon contents. Only in some cases described in the literature [4,5] do the radon activity concentrations measured at the consumers’ faucets turn out to be higher than those measured in groundwater because the distribution systems’ scaling deposits would be capable of adsorbing radium and, upon releasing it, of increasing the radon-in-water activity concentrations just before the point of home water usage [6,7].

All this assumed, to protect the population from health consequences of excessive exposure to natural radiation due to radon, mainly from the risk of cancer onset, it is necessary to investigate the levels of radon in each source including drinking water. 

Many international organizations have introduced some regulations concerning acceptable concentrations of this radionuclide in drinking water.

In Italy, since 2002, when the Italian Standardization Organization (UNI)–Nuclear Energy Commission (UNICEN) appointed a specialized working group, formed by several regional laboratories, national research bodies and Universities, with the main task of writing a set of standards on drinking water measurements, the national institutional authorities have been following with great attention the development of measurement techniques and protocols about the radiological characterization of drinking water, considering the possible health risks to the population.

In this scenario, a series of surveys, as this one, were performed by the authors in the province of Salerno, which started throughout the Italian territory under the initiative of the regional competent authorities in order to assess the health risk of drinking water. In particular, the general approach for controlling radiological hazards was articulated in the following:an initial screening to determine whether the activity concentrations (in Bq/L) are below the reference level at which no further action is required;a second, more accurate, investigation on the concentrations of individual radionuclides only when screening levels are exceeded.

The acceptable levels of radioactivity in water below which no action is required, according the Italian Legislative Decree 28/2016 [8], which is derived from the European Directive 2013/51/ Euratom [9], is 100 Bq/L.

Therefore, the data reported in this paper were compared with the abovementioned reference value and the associated health risk is calculated.

## 2. Study Area

The study area, covering an overall surface extension of about 500 km^2^, is located in the southern part of the Campania region in three distinctive river basins of the Cilento and Diano Valley National Geopark: the Alento, Bussento and Mingardo, from NW to SE.

In this area, 22 springs and wells and related fountains and tanks (see Figure 1 and Table 1) were identified and monitored for the initial screening phase at the district level in 2008–2009 to check that activity concentrations (in Bq/L) were below the defined threshold value of 100 Bq/L.

The water provided by these springs and wells, in fact, is collected for drinking purposes and distributed to 55 municipalities, serving a community of more than 95,000 users. The water management is operated by a joint stock company, Consac water services S.P.A. (Vallo della Lucania, Italy), with total public capital held by the municipalities like shareholders [10].

The CONSAC company was chosen as a pilot aqueduct consortium during the extensive in situ measurement campaigns because it manages a great number of springs (around 60 springs) in the Cilento area. The screening was carried out in the framework of regional research programs, aimed at the investigation and assessment of the impact of radon in natural and anthropic environments on the territory of the Campania region, Southern Italy [11,12,13].

In the present study, 88 water samples from 22 sources were chosen as representative samples of drinking water distributed by the CONSAC aqueduct. All samples were collected at locations where the water is collected by the CONSAC infrastructure, and from there, following all standard treatments for potabilization, the system reaches the final consumers at the territory at the end of the distribution network.

### Hydrogeological Setting

The monitoring system made for monitoring of the springs, the results of which are shown in this paper, covers different hydrogeological structures of the Cilento area. In the following, the main characteristics of these hydrogeological structures are synthetically described.

The Sacred Mount (Gelbison) consists of a conglomeratic and conglomeratic-arenaceous formation, characterized by a mixed permeability, by fractures and porosity and by a medium degree of average permeability. This unit, which extends for about 21 km^2^ and which has a significant annual recharge (annual underground water potential of 7.80 × 10^6^ m^3^/s), is of considerable importance for the drinking water schemes of the Cilento area.

The substructure of the Raia del Pedale belongs to the hydrogeological unit of Mount Cervati-Mount Vesole, which extends from the Diano valley (Vallo di Diano) to the Sele plain and is mainly made up of limestone rocks from the Alburno-Cervati-Pollino stratigraphic-structural unit. At the northwestern and southern margins, there are marked tectonic contacts with poorly permeable or impermeable aquitards.

The Cervati massif constitutes a high hydrogeological zone, in which there are different flow lines of the base aquifer due to the presence of important tectonic discontinuities that divide the carbonate massif into several blocks. In particular, in the central-southern portion, the underground water flow is oriented towards SW to feed the source group of the Fistole del Faraone taped with booty and field wells from CONSAC (about 0.70 m^3^/s annual average).

The substructure of the Southern Cervati presents a multi-aquifer trend, characterized by high springs (la Peta passage), intermediate springs (see Montemezzano) and basal springs, such as the Fistole di Sanza, all captured for drinking purposes by CONSAC. The Fistole di Sanza well field is also located in correspondence with the last spring group.

The structure of Mount Bulgheria consists of dolomites, limestones and marly intercalations. This structure is bounded to the north by the tectonic overlap on the Miocene clayey-marly succession and to the south by the freshwater–seawater interface. It is characterized by a diffuse outflow towards the south with numerous submarine sources with an overall flow rate of less than 100 L/s.

The hydrogeological structure of Mount Forcella Serralunga-Coccovello is divided into different substructures with different flow directions. The substructure of Mount Forcella drains the water underground towards the northeast in correspondence with the spring group of Tredici Fistole, located just upstream of the Sabetta reservoir (average annual flow rate greater than 2 m^3^/s). The substructure of Morigerati, located to the southwest, contains the underground karst system of the middle Bussento (the sinkhole of Caselle in Pittari and the resurgence of Morigerati). It drains both towards the karst aquifer, emerging at the old mill and at the Bussento resurgence, and towards the basal aquifer near the Sicilì bridge. The subsequent hydrogeological substructure of Mount Cocuzzo drains towards the Capello spring, near which the Capello well field (CONSAC well field) is located.

A secondary direction of groundwater flow drains towards the north-northeast direction, emerging at the Melette spring, captured by the CONSAC. The Serralunga-Coccovello substructure, instead, drains towards the Policastro Gulf, both towards the coastal Ruotolo spring (captured in the tunnel by CONSAC) and towards the springs of Vuddu.

## 3. Materials and Methods

### 3.1. Sampling Protocols

As already confirmed by experience in the literature, measurements of radon activity concentrations in different natural water bodies like rivers, springs, groundwater, surface waters and drinking waters on samples are affected by some technical difficulties with respect to radon measurements in other matrices [14,15,16].

Therefore, in order to provide scientifically valid results, it is fundamental to implement a detailed protocol for sampling and measurement [17].

In the framework of the project, internal protocols developed by the Amb.Ra. laboratory (ISO 9001 certified) were proposed for all kinds of radon measurements in different environmental matrices.

Regarding radon-in-water measurements, the most critical factors are the sampling technique and the sealing procedure. Moreover, according to the type of experimental technique used, other factors may include sample concentration, sample size, the sample container, counting time, temperature, relative humidity and background effects.

When the measurement technique is based on the extraction of radon from water to air, as in this case, using alpha spectrometry of the short-lived radon progeny, the sampling technique turns out to be a critical factor and the major source of error in the measurement. In fact, in order for the water sample to be representative of the drinking water analysed, it is extremely important that it has never been in contact with air.

According to the chosen measurement technique for radon in water, described in more detail in the next section, two calibrated volumes of vials were available, 40 mL and 250 mL, provided by the instrumentation manufacturer [18]. The smaller vials are suitable for measuring expected radon-in-water activity concentrations higher than 100 Bq/L, while the larger ones are suitable for concentrations lower than that value.

Because the expected radon-in-water activity concentrations turn out to be lower than 100 Bq/L, only 250-mL vials were used.

Generally, two possible sampling methods can be usually used: preventive and usual [19].

The so-called preventive method aims to obtain a near laminar water flux, avoiding, thus, spontaneous degassing of dissolved gases during the filling operation. This can be achieved by inclining the vial and by reducing the water flux at the minimum value.

Conversely, the typical method is the one that an average consumer would use for filling a vessel, placing it in the vertical position.

All the samples analysed in this paper were collected using the preventive approach.

Inhibition of the radon degassing phenomenon through the material of the vial [18,19] during transportation was avoided by using glass vials and caps of vessels certified by the manufacturer.

Moreover, to further prevent radon degassing from sampling to the analysis phase, each vial was inverted to check for air bubbles during sampling and was always capped with TEFLON lined caps as quickly as possible after filling them up [18,20]. In the presence of air bubbles, the sampling procedure was repeated. Each sample was labelled with a specific code indicating the sampling information (station code, date, time and operator’s name), and the vials were stored in a cooler bag for safe transportation and late analysis in the Amb.Ra. laboratory at the University of Salerno.

### 3.2. The Radon-in-Water Measurement Technique

The radon-in-water measurement technique chosen was alpha spectrometry of the short-lived radon progeny, for its production of a clear-cut physical signal for radon with no contamination from the environment. A calibrated, portable, battery-operated, continuous radon monitor, RAD7 (Durridge Company, Billerica, MA, USA), was used.

In particular, the radon activity concentrations in water samples were determined by using the a RAD7 radon detector coupled with a RAD-H2O kit. Figure 2 shows a schematic diagram of the experimental setups for preparation of the instrument (“purge” phase) before the radon-in-water measurements in order to drop humidity under 6% and to remove the old air by flushing it with fresh air.

The RAD7 was set to a predefined protocol (Wat-250) for a total duration of 30 min and divided into 6 cycles with individual durations of 5 min. In the first cycle, an option which causes bubbling inside the sample is active in order to extract the radon and to enable it to diffuse in the tubes. In the second, there is a resting phase necessary to reach a state of equilibrium within the closed air circuit. The extraction efficiency, i.e., the percentage of radon extracted from the sample water, is very high, typically 94% for a 250 mL vial. In the remaining four cycles (last twenty minutes), the alpha particles produced by decay of the radon progeny are “counted” and a short report is printed for each cycle. The report shows a cumulative spectrum (Figure 3) and, then, ^222^Rn activity concentrations in Bq/L (disintegration per second per litre) expressed with 2*σ* uncertainties. The minimum detection limit claimed by the manufacturer for radon in water is less than 0.37 Bq/L, as found also by other authors [20].

Data acquisition was carried out using the Capture software supplied with the RAD7 detector [22].

### 3.3. Radon Decay Correction

During the radon-in-water analysis or in the case when a water sample is taken and analysed sometime later (rather than immediately), the sample’s radon concentration will decline due to radioactive decay and partly due to the degassing phenomenon, as it has been described before. Hence, it is essential that the resulting activity concentrations should be decay-corrected back from the time the sample was drawn (time of sampling) to the time the sample was analysed. The decay correction is a simple exponential function with a time constant of 132.4 h, coming from the exponential law for radioactive decays. The time elapsed for the sample collection and analysis have to be corrected using the following equation:*C* = *C*_0_·*e*^−*λ·t*^(1)
where *C* is the measured concentration, *C*_0_ is the initial concentration at the sampling time and *t* is the time elapsed since collection (hours). Usually, decay correction is required to correct the radon result back to the sampling time. The decay correction (1) can be used up to 10 days after sampling.

## 4. Results

During the campaigns in Cilento for monitoring the radon activity concentrations occurring in water samples collected from springs captured by the infrastructure of the public company CONSAC to provide drinking water to the population, a total of 84 water samples were collected at the different stations listed in Table 1.

The sampling collection was done for different seasons of the year, according to the local weather conditions, that in some periods have put severe limitations on access to the monitoring stations.

Anytime for the same monitoring station that more measurements were available, an average value was calculated and is reported in Table 2, while only a few of them, 6 exhibiting anomalous large relative errors (more than 10%), were discarded.

All the activity concentrations measured in the laboratory were systematically corrected for radon decay, according to Equation (2), so the values reported in Table 2 all refer to their own sample collection time.

From Table 2 and the histogram drawn in Figure 4, it is evident that the radon activity concentrations in CONSAC drinking water samples vary, with a mean value of 17 Bq/L, from a minimum of 3 Bq/L, measured at a public fountain, to 45 Bq/L, a typical value for groundwater occurring in the hydrogeological units present in the study area, which form a karst landscape and have been already investigated by the authors in other papers. In Figure 4 and Table 2, the radon-in-water activity concentrations measured by the authors in the 22 different typologies of groundwater springs in the province of Salerno are shown.

Finally, all the activity concentration values turn out to be lower than 100 Bq/L (Figure 5), the recommended reference value by the Council Directive 2013/51/Euratom and accepted by the national legislation. The distribution, divided per basin (Figure 5), shows how the Gelbison unit (conglomerate) presents lower values than the Karst unit of Mingardo and Bussento.

### Annual Effective Dose Calculation for Ingestion (H_ing_)

The ingestion of drinking water containing radon concentrations contributes to increased radiation dose exposure, especially in the stomach. The average annual effective dose for ingestion is calculated according the following equation:*H_ing_* = *C_Rn_*·*D_ing_*·*L*,(2)
where *H_ing_* is the annual effective dose measured due to water ingestion and is expressed in (µSv/y), *C_Rn_* is the radon-in-water activity concentration expressed in Bq/L, *D_ing_* is the conversion factor for ingestion due to radon in water (10^−8^ Sv/Bq for adults and 2 × 10^−8^ Sv/Bq for children) and L is the water per capita consumption in a year [14].

In this paper, a per-capita consumption of 730 L in a year (average of 2 L/day) was considered in order to fix the risk of the maximum general consumption according a precautionary principle. Then, also, a pro capita consumption of 50 L and 75 L per year for adults and children, respectively, was defined according to the United Nations Scientific Committee on the Effects of Atomic Radiation (UNSCEAR) [23].

The results (Table 3) show that the annual effective dose values due to ingestion *H_ing_*, according the UNSCEAR method are lower than the recommended reference limit value, which suggest an effective mean dose of 0.1 mSv/y.

The averages of the experimental outcomes from this study, as shown in Figure 6, are 125 µSv/y for adults and 250 µSv for children, clearly lower than the aforementioned threshold.

## 5. Discussion and Conclusions

In this study, some results from the radon-in-water measurement campaigns performed during the screening phase of the research program Rad_Campania carried out by the AmbRa laboratory, University of Salerno and the interuniversity Center for Applied Research on major hazards, CUGRI, to preliminarily assess the radon risk in part of the Campania region are described. In particular, the outcomes refer to the investigations on drinking water samples collected at different capture locations managed by the public company CONSAC operating in the Cilento area of the Salerno province. Data from 88 water samples from the most 22 representative springs from the hydrogeological point of view of the area were elaborated.

The radon-in-water activity concentrations were measured by means of alpha spectrometry of the short-lived radon progeny with a portable, battery-operated, continuous radon monitor.

The results show that the levels of radon in drinking water distributed by CONSAC to more than 95,000 inhabitants in 55 municipalities is well below the reference levels indicated by the EU Council Directive and the current national legislation. Using the knowledge of the materials forming the aqueduct infrastructure, it is possible to argue that the concentrations are significantly lower at delivery point (in-house water tap) or negligible in bottled water, as it is shown by the measurements performed at some fountains (Table 2)

The chosen measurement technique turned out to be very suitable and reliable for the purpose of assessing radon in drinking waters due to its intrinsic characteristics: the physical signal detected (alpha radiation) is not affected by the natural radioactivity background (mainly gamma radiation) and it is distinguishable from the ones associated with the occurrence of other radon’s isotopes, like thoron.

The corresponding annual effective doses by ingestion (0.1 mSv/y) turn out to be fully compliant with the action levels issued by WHO and UNSCEAR. From them, therefore, it can be argued that the water examined is safe for drinking and other domestic purposes, and no remedial actions are demanded.

However, even though these results show low values of radon activity concentrations and doses in perfect compliance with the official institutional indications and so, by the law, the corresponding drinking waters can be considered safe, it has to be underlined that periodical screening could be necessary in order to guarantee the safety of drinking water.

Therefore, more measurements and investigations are needed, not only for progress in scientific knowledge (change in the radon concentrations could be useful to monitor for example change in the hydrogeological units or could identify the problem of scaling deposits in the aqueduct system) but also for the sake of public health [22].

## Figures and Tables

**Figure 1 materials-13-05840-f001:**
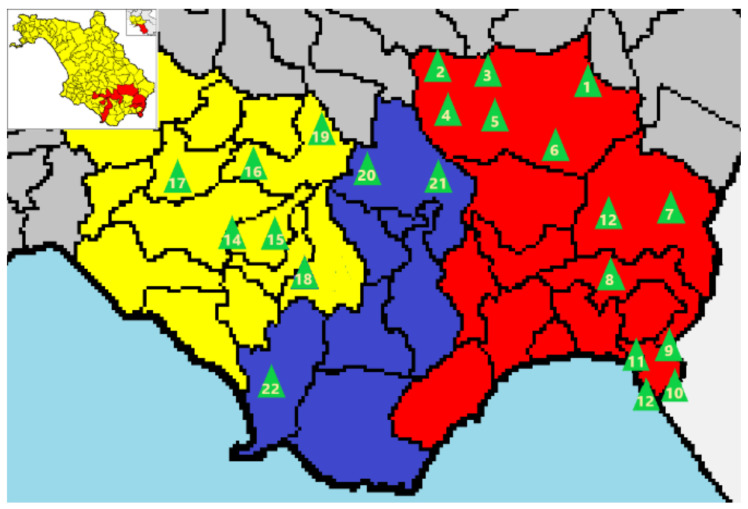
In the upper left-hand corner, shown in yellow is the entire province of Salerno with evidence (in red) of the area under study. In the larger picture, the locations of the 22 groundwater springs and the corresponding drainage basins areas investigated are shown: Bussento (in red), Mingardo (in blue) and Gelbison (in yellow).

**Figure 2 materials-13-05840-f002:**
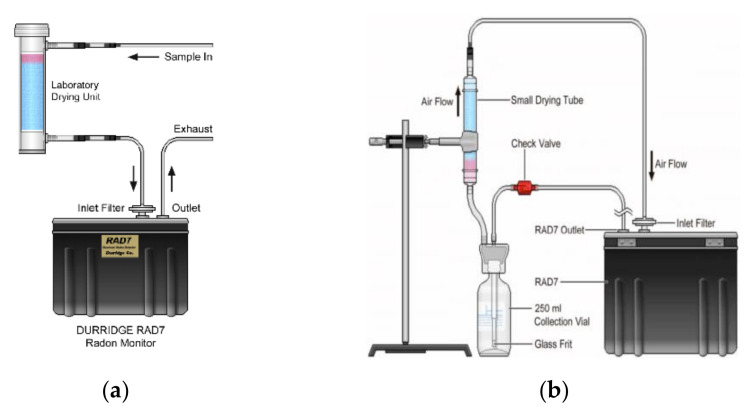
(**a**) Experimental setup for the “purging” phase and (**b**) experimental setup for radon-in-water measurements (modified from [20,21]).

**Figure 3 materials-13-05840-f003:**
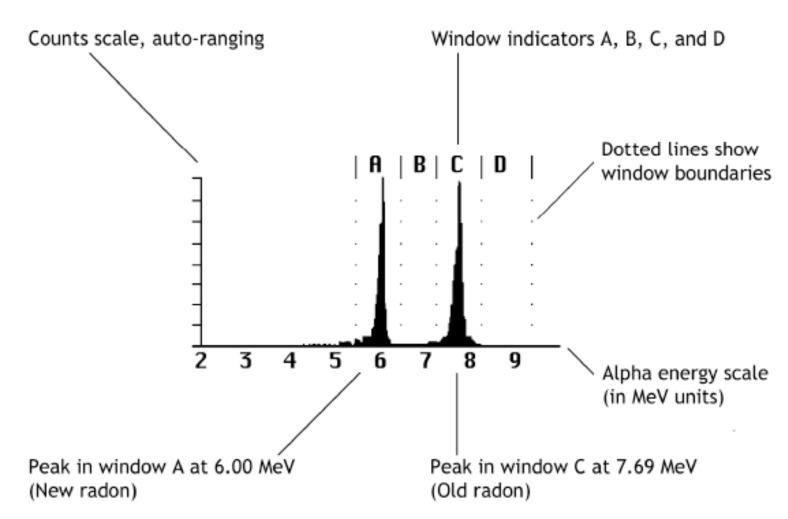
Alpha energy spectrum for alpha-emitting short-lived 222 Rn progeny: 218 Po (peak A) and 214 Po (peak C) at the secular equilibrium (modified from [21]).

**Figure 4 materials-13-05840-f004:**
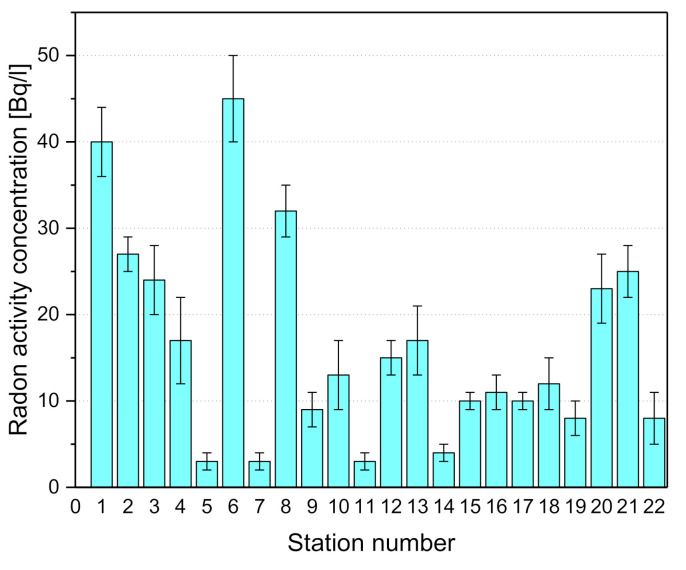
Radon-in-water activity concentrations measured in different typologies of groundwater springs in the province of Salerno, southern Italy.

**Figure 5 materials-13-05840-f005:**
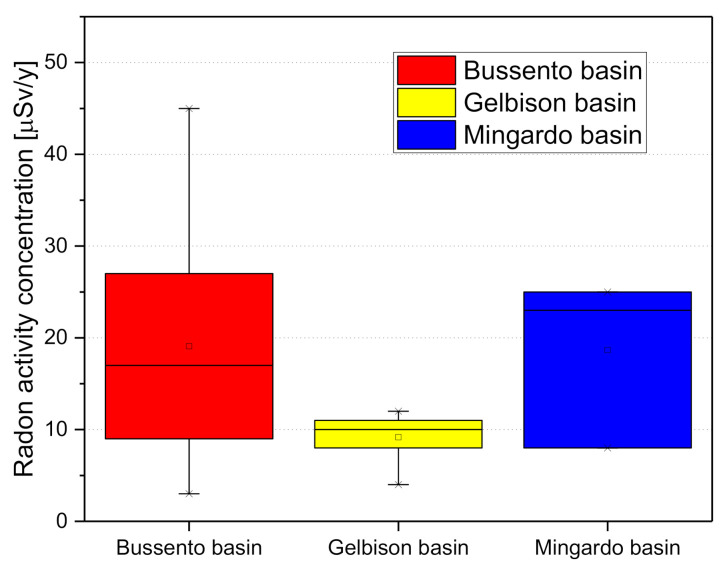
Radon-in-water activity concentrations distribution in the collected samples.

**Figure 6 materials-13-05840-f006:**
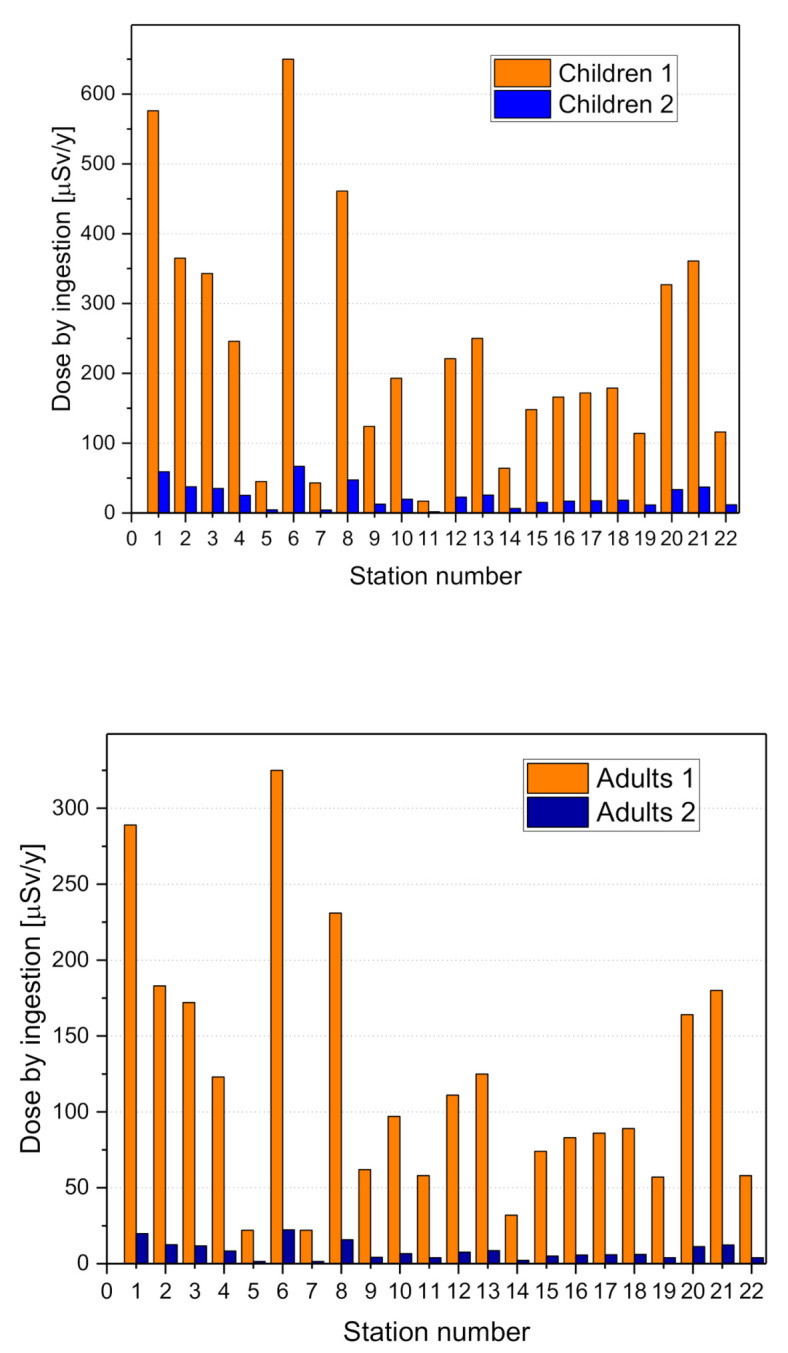
Annual effective dose due to ingestion for adults and children, calculated with a water consumption of 730 L/y (ADULTS1 and CHILDREN1) and with a water consumption of 75 l and 50 l for children and adults respectively according the UNSCEAR2000 (ADULTS2 and CHILDREN2).

**Table 1 materials-13-05840-t001:** Monitoring stations: progressive number, station code, station name and corresponding hydrogeological units. The station code is made of three strings of alphanumerical characters separated by an underscore. The first one represents the water system manager (CONSAC), the second one is the fluvial basin to which the station belongs (BS = Bussento, GLB = Gelbison and MG = Mingardo) and the third group of alphanumerical characters represents the location where the drinking water was sampled (S = spring, F = fountain, P = well and A = reservoir).

n.	Station Code	Station Name	Hydrogeological Unit
1	CONS_BS_S08	Fistole Sanza	Cervati-Karst carbonate
2	CONS_BS_S07	Montemezzano spring	Cervati-Karst carbonate
3	CONS_BS_S07_GS	Montemezzano Geospring	Cervati-Karst carbonate
4	CONS_BS_S07_V	Montemezzano tank	Cervati-Karst carbonate
5	CONS_BS_S07_BIS	La Peta passage	Cervati-Karst carbonate
6	CONS_BS_P03	Sanza well	Cervati-Karst carbonate
7	CONS_BS_ADB_F	Acqua di Battaglia fountain	Tempe di Tortorella
8	CONS_BS_P04	Capello well	Cocuzzo-Serralunga
9	CONS_BS_A01	Sapri reservoir	Serralunga-Coccovello-Karst carbonate
10	CONS_BS_A01_V	Sapri reservoir tank	Serralunga-Coccovello
11	CONS_BS_A01_F	Sapri reservoir fountain	Serralunga-Coccovello
12	CONS_BS_S09	Melette spring	Cocuzzo-Serralunga-Karst carbonate
13	CONS_BS_S10	Ruotolo Coastal spring	Serralunga-Coccovello-Karst carbonate
14	CONS_GLB_S01_P	Elce 1 springs group	Gelbison-Conglomerate
15	CONS_GLB_S02_S	Elce 2 springs group	Gelbison-Conglomerate
16	CONS_GLB_S03	Fiumefreddo spring	Gelbison-Conglomerate
17	CONS_GLB_S04	Palistro springs	Gelbison-Conglomerate
18	CONS_GLB_S05	Fistole Montano	Gelbison-Conglomerate
19	CONS_GLB_VTL	Tempa del Lupo	Gelbison-Conglomerate
20	CONS_MG_S06	Fistole Faraone	Raia del Pedale-Karst carbonate
21	CONS_MG_P01	Faraone wells 1 and 2	Raia del Pedale-Karst carbonate
22	CONS_MG_P02	Mingardo well	Bulgheria-Carbonate

**Table 2 materials-13-05840-t002:** Radon-in-water activity concentrations measured for the different chosen stations.

n.	Station Code	Radon Activity Concentrations (Bq/L)
1	CONS_BS_S08	40 ± 4
2	CONS_BS_S07	27 ± 2
3	CONS_BS_S07_GS	24 ± 4
4	CONS_BS_S07_V	17 ± 4
5	CONS_BS_S07_BIS	3 ± 1
6	CONS_BS_P03	45 ± 5
7	CONS_ADB_F	3 ± 1
8	CONS_BS_P04	32 ± 3
9	CONS_BS_A01	9 ± 2
10	CONS_BS_A01_V	13 ± 4
11	CONS_BS_A01_F	3 ± 1
12	CONS_BS_S09	15 ± 2
13	CONS_BS_S10	17 ± 4
14	CONS_GLB_S01_P	4 ± 1
15	CONS_GLB_S02_S	10 ± 1
16	CONS_GLB_S03	11 ± 2
17	CONS_GLB_S04	10 ± 1
18	CONS_GLB_S05	12 ± 3
19	CONS_GLB_VTL	8 ± 2
20	CONS_MG_S06	23 ± 4
21	CONS_MG_P01	25 ± 3
22	CONS_MG_P02	8 ± 3

**Table 3 materials-13-05840-t003:** Related effective annual dose due to ingestion for adults and children according to ^1^ the maximum annual consumption and according to ^2^ the UNSCEAR medium annual consumption.

n.	Station Code	Adults ^1^ (μSv/y)	Children ^1^ (μSv/y)	Adults ^2^ (μSv/y)	Children ^2^ (μSv/y)
1	CONS_BS_S08	289	576	20	59
2	CONS_BS_S07	183	365	13	38
3	CONS_BS_S07_GS	172	343	12	35
4	CONS_BS_S07_V	123	246	8	25
5	CONS_BS_S07_BIS	22	45	2	5
6	CONS_BS_P03	325	650	22	67
7	CONS_ADB_F	22	43	2	4
8	CONS_BS_P04	231	461	16	47
9	CONS_BS_A01	62	124	4	13
10	CONS_BS_A01_V	97	193	7	20
11	CONS_BS_A01_F	58	17	4	2
12	CONS_BS_S09	111	221	8	23
13	CONS_BS_S10	125	250	9	26
14	CONS_GLB_S01_P	32	64	2	7
15	CONS_GLB_S02_S	74	148	5	15
16	CONS_GLB_S03	83	166	6	17
17	CONS_GLB_S04	86	172	6	18
18	CONS_GLB_S05	89	179	6	18
19	CONS_GLB_VTL	57	114	4	12
20	CONS_MG_S06	164	327	11	34
21	CONS_MG_P01	180	361	12	37
22	CONS_MG_P02	58	116	4	12

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
