# Peer review of "Alpha Spectrometry of Radon Short-Lived Progeny in Drinking Water and Assessment of the Public Effective Dose: Results from the Cilento Area, Province of Salerno, Southern Italy"

_materials, 2020, doi:10.3390/ma13245840_

Round 1
Reviewer 1 Report
1005741
GENERAL COMMENTS
It is important to study radon and radium, no question. But it is unclear why the study was done in the first place-was there a reason to focus on radon in these springs noted in the manuscript (were they suspected of having high radon concentrations?).
The writing quality in the document is such that it takes away from the reader’s ability to concentrate on the content of the document. Writing is often repetitive (even within the same sentence), uses double negatives (e.g. not negligible, Abstract line 21), and many paragraphs (e.g. the first three in the Introduction section) consist only of one or two sentences, which is abnormal, and the word order in many sentences is incorrect making the writing difficult to decipher at times (see below for one example).
There are a number of strange words in the document that should be double-checked to make sure the authors really intend to use them. These may be examples of “lost in translation.” For example:
Fistula appears 7 times in the document; to my knowledge this word (in English, anyway) is almost exclusively used in a medical sense and not in the context of geology; maybe there are other more suitable terms for describing interconnected subsurface channels in karst topography?
Booty well (page 4, line 150)—booty in English means treasure or rear end
Text and labels in most of the figures are nearly illegible; labels in figures 1 & 2 are impossible to read. The image quality in the figures is poor; the authors need to find a way to preserve the resolution of the original images if they wish the readers to be able to read the labels in maps or on graphs.
ABSTRACT
The abstract states that a study was done to investigate radon levels in the water of 22 springs in the vicinity of Cilento, Italy, and none of the springs contained radon at levels above 0.1 mSv/y. The authors refer to this level as the “global average” but in fact it is a recommended reference dose level (RDL) which is equal to 10% of the level at which an intervention would be necessary to reduce health risk. The authors also state that since the radon levels were below 0.1 mSv/y that the exposure was negligible, but whether an exposure below the RDL of 0.1 mSv/y constitutes negligible health risk is unclear.
The abstract states in the second sentence that radium is everywhere-if by everywhere the authors mean in minute concentrations, then yes. The distribution of radium throughout the earth is far from uniform, though!
INTRODUCTION
This section contains some of the information that would be expected in this section, but it lacks polish and is incomplete. If the authors wish to begin with defining what safe drinking water means, that is fine, but for example, please explain what organizations and laws or policies govern drinking water. And please take care that the statements in the introduction actually make sense. For example, it is important to consider dose and ‘activity’ in the case of radon in drinking water. But the authors seem to state that the dose in the case of non-radiological contaminants is unimportant—this is, however, untrue.
Much of the writing makes sense if one pauses to think about how the sentences should be worded, but there is a lot of work to do to polish this manuscript enough to be easily readable. For example, in lines 53-4:
What is being produced or generated and how?
Rocky body: meaning rock?
“fluid carriers…” meaning water?
A similar number of questions could be asked about many of the sentences in the introduction.
MATERIALS AND METHODS
Writing in the section is particularly difficult to follow and uses extremely general language and overall way too many words to state that the sample collection procedure is important because it may affect results, particularly in the case of water to air extraction. The difference between typical and preventive sample collection is important, but the authors also use the word “usual” and should choose either usual or typical, but not both.
Word order in sentences is also a problem, leading to significantly different meanings than the authors likely intended. See for example lines 216-217 where it is suggested that “…the transportation procedure can be strongly influenced by the radon degassing phenomenon…”—but it is likely that the authors meant that degassing can be strongly influenced by transportation? Is it really transportation or actually sample storage that the authors are concerned with (the discussion appears to be about vessel material and cap).
RESULTS
In the materials and methods section, the study description indicates 78 water samples in 22 locations…how were the locations (why these particular springs??) determined and how many samples from each location? In the Results section, it states that 84 samples were collected—which is it, 78 or 84? Also, the data are 12 years old. Why are they just now being published?
Lines 304-306: this explanation of when and why samples were taken is insufficient. What are the weather problems that cause samples not to be taken. When were the samples taken?
Lines 307-309: what was the decision point (threshold) for discarding data that were deemed to be in error?
CONCLUSIONS
Lines 381-384 make no sense as a conclusion since the whole manuscript is about health concerns and the authors at the end of the manuscript suddenly turn to discussing potential benefits of radiation exposure?
The final conclusion (lines 385-6) also makes little sense. The authors have not made a case that radon doses in the study area are in any way significant enough to present a public health concern, and the claim that small doses of radiation may be beneficial is supported only by an obscure document from an international youth science conference.
Reviewer 2 Report
The article titeled "Alpha spectrometry of radon
short-lived progeny in drinking waters and assessment
of the public effective dose. Results from the Cilento area,
province of Salerno, southern Italy" submitted by
Enver Faella et al. presents Rn-222 data for water samples
taken in Southern Italy.
General comments
Sampling and measurement are described in full detail.
Whereas the sampling procedure may be interesting for
others, details about the measurement don't give
any new information. The method described is just
state of the art. Should be deleted.
There is some confusion about what ingestion dose
from Rn-222 has to be respected.The Italian Legislative
Decree 28/2016 clearly states a reference level of
0.1 mSv/y, in accordance with the EU Directive 2013/51.
But at line 348ff the authors mention a reference dose
of 1 mSv/y, comparing this with their calculated doses.
In addition their dose calculations are based on not
justified water consumption. They are using a per
capita consumption of 730 L/y for adults and children.
In the UNSCAR 1993 report, Annex A, p 54 far lower
consumption rates are used to calculate the ingestion doses,
50 L/y for adults, 75 L/y for children and 100 L/y for infants.
Citation : "Since radon is readily lost by heating and bottling,
the consumption of interest here is that of water directly from
the tab".
Should be recalculated with the UNSCEAR 1993 data.
For an Earth-Sciences article the sort of referencing
with numbers in brackets is not standard.
Specific comments
Fig.1 : bad resolution, Codes for the sites not readable.
Figure caption talks about "22 groundwater springs"
but at lines 234/235 one can see that there are also
samples from fountains, wells and reservoirs.
Line 317 : wrong reference [10]
Fig.4 : where are these data from ? Reference should be
given in figure caption.
Table 2 and Fig.6 : as mentioned above the dose rates are calculated
using a far too high drinking water consumption.
Round 2
Reviewer 1 Report
The manuscript is much improved though I recommend additional English language editing prior to publication.
In Figure 1b the codes are still illegible-it looks like an aliasing problem. If the codes are important for the audience, authors should take care to use a different image format (e.g. .png) that will better preserve text legibility --this is also a problem in Figures 5 and 6.
There are still several instances of the word fistula being used in the document. I am not opposed to retaining local flavor of geological names but as the authors note in their cover letter, this is confusing to readers.
Author Response
Please, see the attachment.
